# Neutralization of hepatitis B virus with vaccine-escape mutations by hepatitis B vaccine with large-HBs antigen

Ayaka Washizaki[1,12], Asako Murayama[2,12], Megumi Murata[1], Tomoko Kiyohara[2], Keigo Yato[2], Norie Yamada[2], Hussein Hassan Aly[2], Tomohisa Tanaka[3], Kohji Moriishi [3,4], Hironori Nishitsuji[5], Kunitada Shimotohno[6], Yasumasa Goh[7], Ken J. Ishii [8], Hiroshi Yotsuyanagi[9], Masamichi Muramatsu[2], Koji Ishii[10], Yoshimasa Takahashi [11], Ryosuke Suzuki[2], Hirofumi Akari [1] ✉ & Takanobu Kato [2] ✉

Although the current hepatitis B (HB) vaccine comprising small-HBs antigen (Ag) is potent and safe, attenuated prophylaxis against hepatitis B virus (HBV) with vaccine-escape mutations (VEMs) has been reported. We investigate an HB vaccine consisting of large-HBsAg that overcomes the shortcomings of the current HB vaccine. Yeast-derived large-HBsAg is immunized into rhesus macaques, and the neutralizing activities of the induced antibodies are compared with those of the current HB vaccine. Although the antibodies induced by the current HB vaccine cannot prevent HBV infection with VEMs, the large-HBsAg vaccine-induced antibodies neutralize those infections. The HBV genotypes that exhibited attenuated neutralization via these vaccines are different. Here, we show that the HB vaccine consisting of large-HBsAg is useful to compensate for the shortcomings of the current HB vaccine. The combined use of these HB vaccines may induce antibodies that can neutralize HBV strains with VEMs or multiple HBV genotypes.

Hepatitis B virus (HBV) infection leads to chronic liver diseases, such as chronic hepatitis, cirrhosis, and hepatocellular carcinoma[1,2]. In 2015, the World Health Organization estimated that approximately 257 million people worldwide were infected with HBV and at risk for cirrhosis and hepatocellular carcinoma[3]. Mother-to-child transmission is a critical route of transmission, but other transmission routes in adults, such as through sexual contact or intravenous drug use, also contribute to the global spread of HBV because HBV in blood or body fluids is highly contagious[4,5]. Current treatment strategies for chronic hepatitis B (HB) based on nucleos(t)ide analogs and interferons are intended to suppress virus propagation and disease progression, but they do not eradicate the virus in patients[2,6]. Although several trials have been conducted to establish an effective treatment to eliminate HBV from infected patients, this goal has not yet been achieved. Therefore, the HB

[1]Center for the Evolutionary Origins of Human Behavior, Kyoto University, Aichi, Japan. [2]Department of Virology II, National Institute of Infectious Diseases, Tokyo, Japan. [3]Department of Microbiology, Graduate School of Medicine, University of Yamanashi, Yamanashi, Japan. [4]Division of Hepatitis Virology, Institute for Genetic Medicine, Hokkaido University, Hokkaido, Japan. [5]Department of Virology and Parasitology, Fujita Health University School of Medicine, Aichi, Japan. [6]Genome Medical Sciences Project, National Center for Global Health and Medicine, Chiba, Japan. [7]Research Laboratory, Beacle, Inc., Kyoto, Japan. [8]Division of Vaccine Science, Department of Microbiology and Immunology, The Institute of Medical Science, The University of Tokyo, Tokyo, Japan. [9]Division of Infectious Diseases, Advanced Clinical Research Center, The Institute of Medical Science, The University of Tokyo, Tokyo, Japan. [10]Department of Quality Assurance, Radiation Safety, and Information System, National Institute of Infectious Diseases, Tokyo, Japan. [11]Research Center for Drug and Vaccine Development, National Institute of Infectious Diseases, Tokyo, Japan. [12]These authors contributed equally: Ayaka Washizaki, Asako Murayama. ✉e-mail: akari.hirofumi.5z@kyoto-u.ac.jp; takato@niid.go.jp

vaccine is recognized as the most effective approach to control the spread of HBV and reduce HBV-related morbidity and mortality.

HBV has a partially double-stranded 3.2 kb DNA genome and possesses four open reading frames encoding hepatitis B surface (HBs) antigen (Ag), hepatitis B e (HBe) Ag/hepatitis B core (HBc) Ag, hepatitis B polymerase, and hepatitis B X protein in its genome[7]. HBsAg is produced in three forms: small- (S-), middle- (M-), and large- (L-) HBsAg. S-HBsAg has 226 amino acids (aa), while M- and L-HBsAg have additional preS2 (55 aa) and preS1+preS2 (174 aa) regions attached to S-HBsAg, respectively. These HBsAgs are known to form both infectious HBV particles and noninfectious subviral particles; HBV particles consist of all HBsAg species, and subviral particles consist of S-HBsAg alone or in combination with M-HBsAg. L-HBsAg is essential in the formation of infectious viral particles because it contains the receptor-binding domain (RBD) in the preS1 region. Yeast-derived S-HBsAg has been used as the HB vaccine worldwide. Two HB vaccines that employ the yeast-derived S-HBsAg of different HBV genotypes are currently in use in Japan. Both are considered to have similarly strong abilities to induce neutralizing antibodies with a tolerable safety profile. However, several concerns that cannot be neglected have been noted. Approximately 10% of vaccinated adults are known to have a low or no humoral response to these vaccines[8]. In addition, HBV variants with amino acid polymorphisms in the antigenic region (*a* determinant) of S-HBs may infect even vaccinated individuals[9–15]. The amino acid polymorphisms in these variants are considered responsible for the evasion of the virus from neutralization by the antibodies induced by HB vaccines and are designated vaccine-escape mutations (VEMs), although a precise and quantitative evaluation of the effects of these polymorphisms on the neutralizing abilities of HB vaccine-induced antibodies has not yet been achieved.

In this work, to address the concerns regarding the current HB vaccine, we aimed to establish an HB vaccine comprising yeast-derived L-HBsAg. This L-HBsAg was generated by embedding the protein of whole HBs regions in a unilamellar liposome that displayed the regions from preS1 to S-HBs on the surface of the hollow nanoparticles by forming a three-dimensional structure mimicking the HBV particle[16–18]. The preS1 region included in this antigen contained the RBD and is expected to be a promising candidate for the HB vaccine[19,20]. To assess the efficacy and safety of the L-HBsAg vaccine, we combined it with two kinds of effective adjuvants for immunization in rhesus macaques and evaluated the induction of antibody to various regions of HBsAg. To investigate the neutralizing ability of induced antibodies, we exploited the infection system of cell culture-generated HBV (HBVcc) and HBV reporter viruses[21–23]. The HBV reporter virus infection system can be used to evaluate neutralization by vaccine-induced antibodies against HBV variants with VEM or HBV clones of various genotypes in cell culture.

## Results

### Induction of anti-HBs antibodies in rhesus macaques
To assess the immunogenicity of the HB vaccine consisting of yeast-derived L-HBsAg, L-HBsAg (3.0 μg) of the HBV genotype (GT) C was mixed with K3-SPG (100 μg, L-HBs+K3-SPG) or Addavax (500 μL, L-HBs+Addavax) and used to immunize rhesus macaques (*n* = 3 in each group). As a control, the commercially available HB vaccine comprising yeast-derived S-HBsAg of GTC (S-HBs vaccine) was also used to immunize rhesus macaques (*n* = 3). These vaccines were administered three times subcutaneously at 4 and 20 weeks after the initial vaccination following the protocol of the three-dose HB vaccine series in humans, and the observation was continued until 26 weeks after the initial vaccination (Fig. 1). During immunization with vaccines, vaccine-associated clinical abnormalities or severe increases in liver enzyme levels, including levels of alanine aminotransferase, aspartate aminotransferase, and gamma-glutamyl transpeptidase, were not observed (Supplementary Fig. 1).

Regarding immunization with the S-HBs vaccine, potent induction of the anti-HBs antibodies (indicated as anti-S-HBs antibodies in this study to distinguish them from antibodies to other HBsAgs) was detected after the third administration to the macaques, whereas the induction of the anti-S-HBs antibodies by immunization with L-HBs+K3-SPG and L-HBs+Addavax was limited (Fig. 1a). At the endpoint of the observation period, the anti-S-HBs titers after immunization with L-HBs+K3-SPG and L-HBs+Addavax were approximately 53- and 11-fold lower than those obtained with the S-HBs vaccine, respectively (Fig. 1a, right panel). However, potent induction of the anti-L-HBs antibody was detected by immunization with L-HBs+Addavax and was higher than that achieved with the S-HBs vaccine and L-HBs+K3-SPG (Fig. 1b). At the endpoint, the anti-L-HBs titer induced by immunization with L-HBs+Addavax was approximately 3- and 4-fold higher than those induced by the S-HBs vaccine and L-HBs+K3-SPG, respectively (Fig. 1b, right panel). The induction of the anti-preS1 antibody by immunization with L-HBs+Addavax was approximately 17-fold higher than that with L-HBs+K3-SPG, and no induction of the anti-preS1 antibody was detected by immunization with the S-HBs vaccine (Fig. 1c).

### Neutralization of HBV infection by induced antibodies
To assess the neutralizing activity of the induced antibodies, the HBVcc infection system was used (Supplementary Fig. 2a). The 10- or 100-fold diluted serum samples of the immunized macaques obtained at the endpoint of the observation period were mixed with HBVcc (GTC) at 1000 genome equivalents (GEq)/cell and inoculated into HepG2-NTCPsec+ cells[24]. When infection was performed with HBVcc treated with 100-fold diluted serum samples from the S-HBs vaccine-immunized macaques, substantially lower levels of HBc-positive cells were detected than when infection was performed with vehicle-treated HBVcc. After treatment with the 10-fold dilution, HBc-positive cells were scarcely detected, indicating neutralization by the serum samples of the S-HBs vaccine-immunized macaques (Fig. 2a). The number of HBc-positive cells was higher by viral infection after treatment with serum samples from L-HBs+K3-SPG-immunized macaques and was lower by viral infection after treatment with serum samples from L-HBs+Addavax-immunized macaques than with those from S-HBs vaccine-immunized macaques (Fig. 2b).

### Quantification of neutralizing activities of induced antibodies
To precisely quantify the neutralizing effects, the HBV reporter virus (HBV/NL) infection system was used (Supplementary Fig. 2b). The HBV/NL used in this experiment was generated with the HBV GTC strain and designated HBV/NL-wild type (WT). The serum samples of the HB vaccine-immunized macaques were serially diluted in 10-fold increments and mixed and incubated with HBV/NL-WT. Afterward, macaque serum-treated viruses were used to infect G2/NT18-C cells[25]. The HBV infection efficiencies were quantified by the measurement of NanoLuc luciferase (NL) activity in infected cells (Fig. 3a, left panel). The NL activities after HBV/NL-WT infection increased with the dilution factor of the serum samples. The 50% inhibitory concentration (IC$_{50}$) of the serum samples of L-HBs+Addavax-immunized macaques was the lowest ($8.15 \times 10^{-3} \pm 5.25 \times 10^{-3}$-fold), followed by that of S-HBs vaccine-immunized macaques ($1.43 \times 10^{-2} \pm 6.93 \times 10^{-3}$-fold). The IC$_{50}$ value of the serum samples of L-HBs+K3-SPG-immunized macaques was the highest, at $1.37 \times 10^{-1} \pm 2.15 \times 10^{-2}$-fold (Fig. 3a, right panel). The preimmune serum samples of these animals indicated no inhibitory effect of HBV/NL-WT infection (Supplementary Fig. 3). Hereafter, we focused on comparing immunization by the S-HBs vaccine and L-HBs+Addavax.

To confirm whether the observed neutralizing effects depended on the antibodies in serum, the antibodies in serum samples were purified, and their neutralizing effects were investigated. The protein concentrations of antibodies purified from S-HBs vaccine- and L-HBVs+Addavax-immunized macaques were comparable, while those

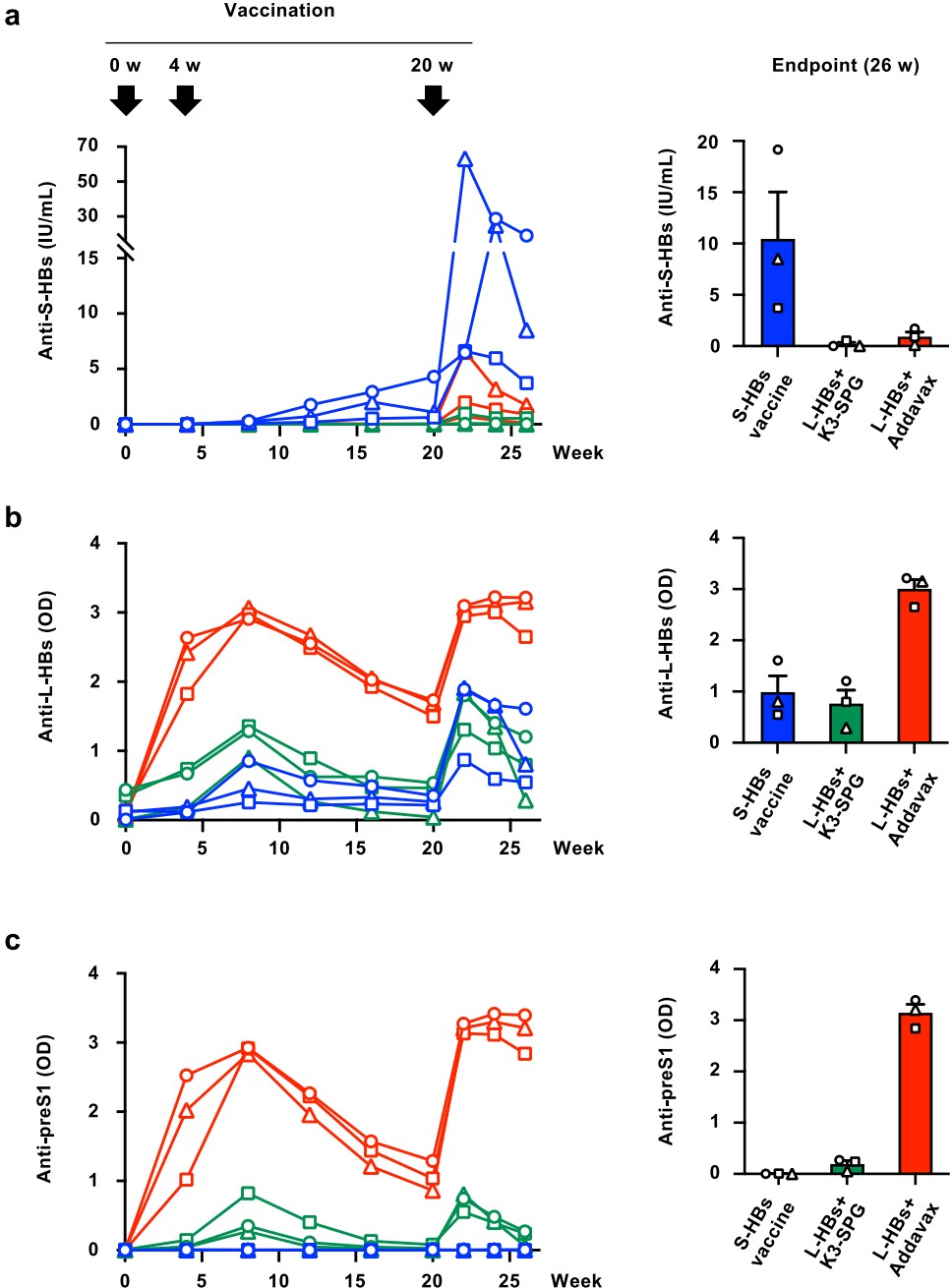

**Fig. 1 | Immunization schedule of HB vaccines and induction of anti-HBs antibodies.** The induction of antibodies to S-HBs (**a**), L-HBs (**b**), and preS1 (**c**) was measured after immunization of rhesus macaques with S-HBs vaccine (blue), L-HBs +K3-SPG (green), or L-HBs+Addavax (red). These vaccines were administered three times subcutaneously at 0, 4, and 20 weeks, and the observation was continued until 26 weeks after the initial vaccination. Blood samples were collected every 4 weeks during the vaccination and every 2 weeks after the last vaccination. **a** The titer of anti-S-HBs antibodies was measured using an Enzygnost Anti-HBs II. **b** The titer of anti-L-HBs antibodies was measured by enzyme-linked immunosorbent assay. The *y*-axis indicates the absorbance at 450 nm. **c** The titer of the anti-preS1 antibody was measured by enzyme-linked immunosorbent assay. The *y*-axis indicates the absorbance at 450 nm. The mean ± SE of the titers of three animals at the endpoint of the observation period is indicated in the right panel. Source data are provided as a Source Data file.

purified from L-HBVs+K3-SPG-immunized macaques were slightly lower than those (Supplementary Fig. 4). Similar to the experiments with serum samples, HBV/NL-WT was incubated with purified antibodies after dilution in 10-fold increments and infected. As expected, treatment with purified antibody resulted in a similar dose–response curve to that observed with serum treatment (Fig. 3b, left panel). The IC$_{50}$ values of purified antibodies of S-HBs vaccine- and L-HBs+Addavax-immunized macaques were $1.01 \times 10^{-2} \pm 4.32 \times 10^{-3}$-fold and $5.45 \times 10^{-3} \pm 3.54 \times 10^{-3}$-fold, respectively (Fig. 3b, right panel). When comparing the results of assays performed with serum samples and

purified antibodies, the ratio of IC$_{50}$ values between S-HBs vaccine- and L-HBs+Addavax-immunization was almost identical.

An antibody adsorption assay of induced antibodies was also conducted. The purified antibodies were prepared at a 10-fold concentration of IC$_{50}$ values and mixed with serially diluted S-HBs or L-HBs antigens. HBV/NL-WT was incubated with antigen-adsorbed antibodies and infected into G2/NT18-C cells. The neutralizing ability of antibodies of S-HBs vaccine-immunized macaques was diminished by incubation with the S-HBs antigen (Fig. 3c, left panel). In contrast, the neutralizing effects of the antibodies from L-HBs+Addavax-immunized

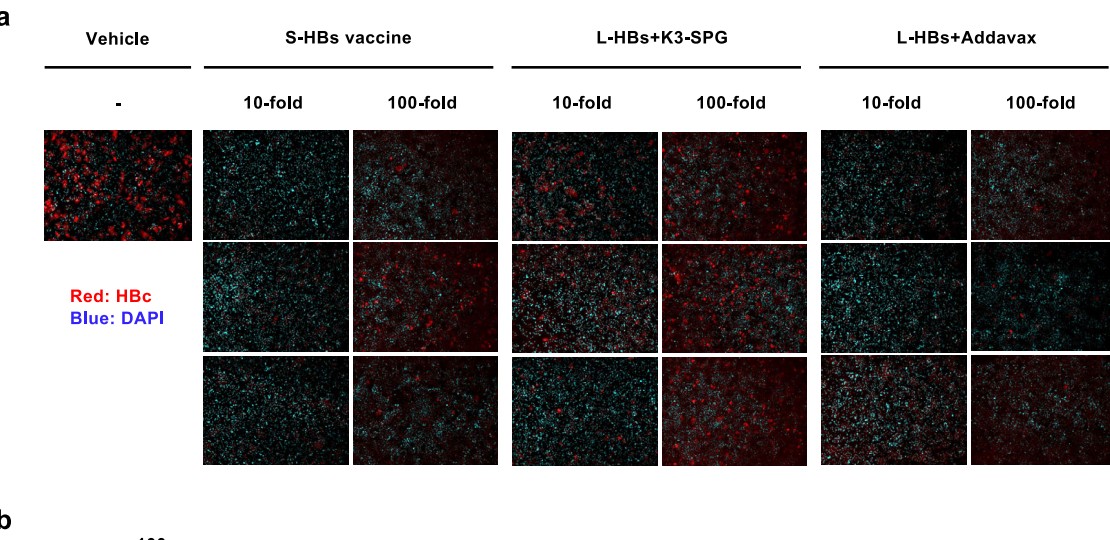

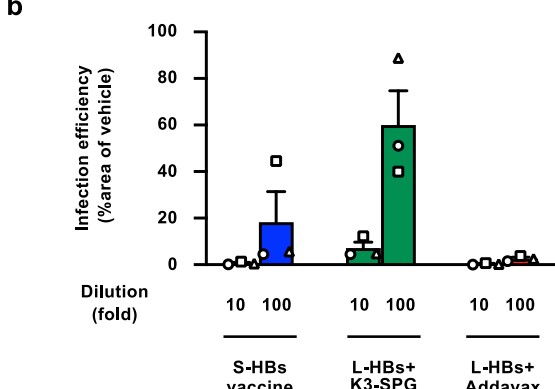

**Fig. 2 | Neutralization of HBVcc infection by serum samples of immunized rhesus macaques. a** HBVcc (GTC, 1000 GEq/cell) was treated with 10- or 100-fold diluted serum samples of rhesus macaques immunized with the indicated HB vaccines and infected with HepG2-NTCPsec+ cells. HBVcc treated with PBS was used as a vehicle control. The infected cells were detected by staining with an anti-HBc antibody 12 days after infection, and the nuclei were visualized by DAPI. The data of three animals in each group are indicated. **b** The infection efficiencies of HBVcc after treatment with plasma samples of rhesus macaques were evaluated. The relative fluorescence intensities of HBc-positive cells in each image were measured and indicated as the percentages of the vehicle treatment. The mean ± SE of three animals in each group is indicated. Source data are provided as a Source Data file.

macaques were rarely affected by incubation with the S-HBs protein. However, the neutralizing effects of antibodies from both S-HBs vaccine- and L-HBs+Addavax-immunized macaques were effectively decreased by incubation with the L-HBs antigen (Fig. 3c, right panel).

### Neutralizing activities of HB vaccine-induced antibodies to VEMs

We then assessed the effects of VEMs on the neutralizing activities of the antibodies induced by HB vaccines using HBV/NL with VEMs. Representative VEMs, including isoleucine to serine at amino acid 126 (I126S), glycine to arginine at amino acid 145 (G145R), and glycine to alanine at amino acid 145 (G145A), were introduced to HBV/NL-WT, and HBV/NL-I126S, HBV/NL-G145R, and HBV/NL-G145A were generated (Supplementary Figs. 2b and 5). These variants of HBV/NL were incubated with serially diluted antibodies and used to infect G2/NT18-C cells. The neutralizing activity of S-HBs vaccine-induced antibodies to HBV/NL-I126S infection was slightly lower (rightward migration of the dose–response curves) than that of HBV/NL-WT infection (Fig. 4a, left panel; refer to Fig. 3b, left panel). The IC$_{50}$ value of HBV/NL-I126S was 1.19 ± 0.0240-fold that of HBV/NL-WT (Fig. 4b and Supplementary Table 1). In experiments using HBV/NL-G145R and -G145A, increased IC$_{50}$ values were detected (Fig. 4a, center and right panel). The IC$_{50}$ values of HBV/NL-G145R and -G145A were 8.45 ± 1.84- and 3.23 ± 0.363-fold that of HBV/NL-WT, respectively (Fig. 4b and Supplementary Table 1). In contrast, the neutralizing activities of L-HBs+Addavax-induced antibodies were not affected by these VEMs. The IC$_{50}$ values of

HBV/NL-I126S, -G145R, and -G145A were similar to those of HBV/NL-WT (Fig. 4b and Supplementary Table 1).

To confirm whether this system can be used to detect the effects of VEMs on neutralization by HB vaccine-induced antibodies for HBV infection, we assessed the neutralizing activities of S-HBs vaccine-induced antibodies in humans. The antibodies were purified from the serum samples of S-HBs vaccine-immunized individuals, and their neutralizing activities against HBV/NL-WT or -G145R were compared. Similar to the antibodies induced in rhesus macaques, the neutralizing activities of S-HBs vaccine-induced antibodies in humans were also attenuated by introducing G145R. The dose–response curve of HBV/NL-G145R exhibited rightward migration from that of HBV/NL-WT, and the calculated IC$_{50}$ value of antibodies to HBV/NL-G145R was 3.89 ± 0.622-fold that of HBV/NL-WT (Supplementary Fig. 6).

### Neutralizing activities against HBV strains of various genotypes

To investigate the genotype dependency of the neutralizing activities of HB vaccine-induced antibodies, HBV/NLs harboring the L-HBs region of GTA (HBV/NL-GTA), GTB (HBV/NL-GTB), and GTD (HBV/NL-GTD) were generated and used (Supplementary Figs. 2b and 5). These HBV/NL strains were incubated with serially diluted antibodies and used to infect G2/NT18-C cells. The neutralizing activities of S-HBs vaccine-induced antibodies to HBV/NL-GTA and -GTB were comparable to those to HBV/NL-WT (GTC) (Fig. 5a, left and center panels; refer to Fig. 3b, left panel). The IC$_{50}$ values of antibodies to these genotypes were 1.31 ± 0.208- and 1.26 ± 0.0876-fold those to HBV/NL-WT (GTC),

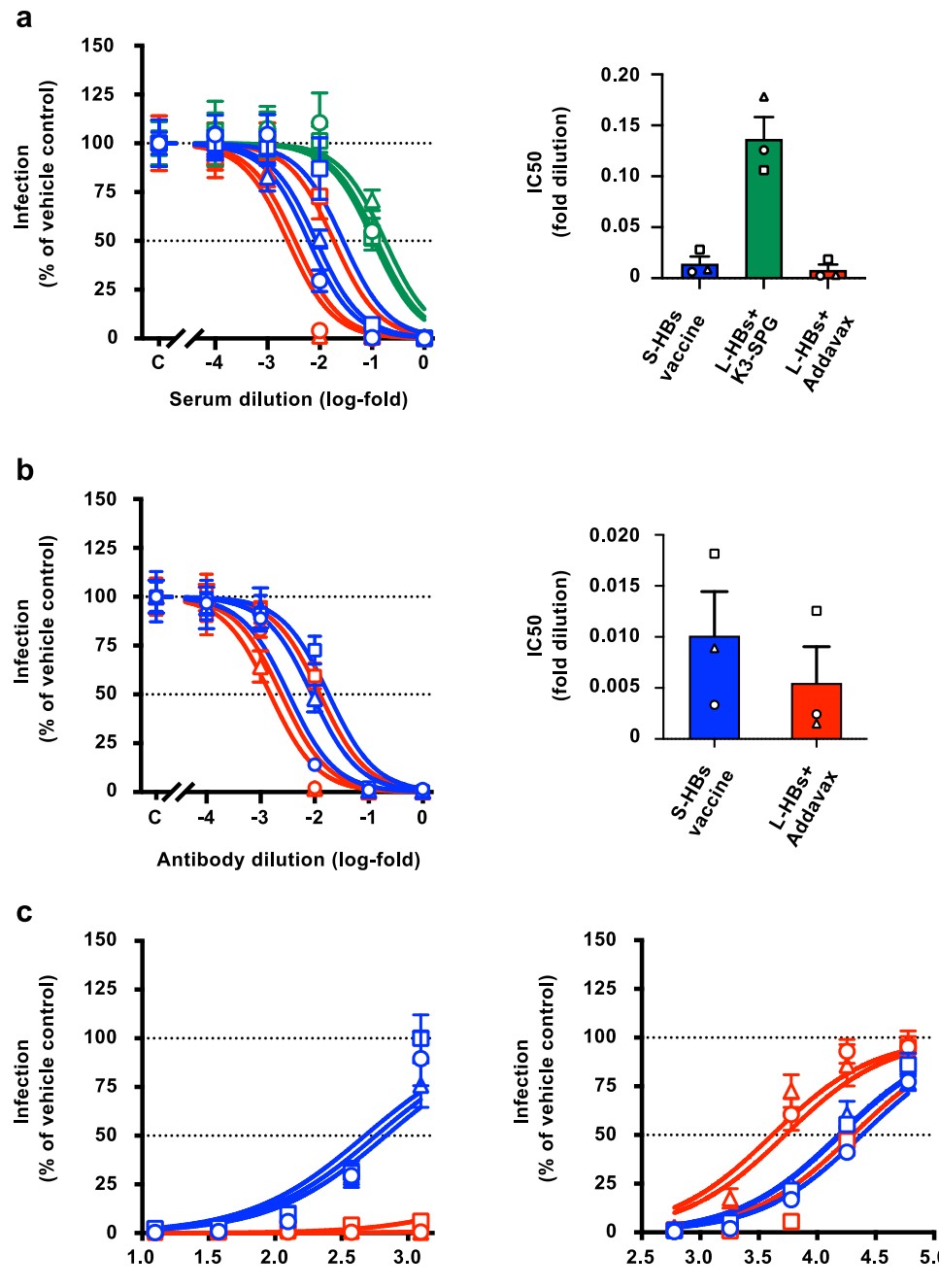

**Fig. 3 | Neutralizing activities in the HBV/NL infection system. a** The neutralizing activities of serum samples were assessed in the HBV/NL-WT infection system. The cell culture-generated HBV/NL-WT (GTC, 20 GEq/cell) was mixed with serially diluted serum samples of rhesus macaques immunized with S-HBs vaccine (blue), L-HBs+K3-SPG (green), or L-HBs+Addavax (red) for 1 h at 37 °C and infected into G2/NT18-C cells. The infection efficiencies at the indicated dilutions of serum were assessed by measuring the NL activities of infected cells in 5 wells at 7 days after infection and the mean ± SD was indicated at each dilution. The dose–response curves with the log (inhibitor) vs. normalized response model were drawn, and the $IC_{50}$ values were calculated. The mean ± SE of $IC_{50}$ values of three animals is indicated in the right panel. **b** The neutralizing activities of purified antibodies were assessed in the HBV/NL-WT infection system. HBV/NL-WT (GTC, 20 GEq/cell) was mixed with serially diluted antibodies induced in S-HBs vaccine- (blue) or L-HBs+Addavax- (red) immunized macaques for 1 h at 37 °C and was used to infect G2/NT18-C cells. The infection efficiencies at the indicated dilutions of antibodies were

assessed by measuring the NL activities of infected cells in 5 wells at 7 days after infection and indicated the mean ± SD at each dilution. The dose–response curves with the log (inhibitor) vs. normalized response model were drawn, and the $IC_{50}$ values were calculated. The mean ± SE of $IC_{50}$ values of three animals is indicated in the right panel. **c** The specificities of induced antibodies to the neutralizing effects were assessed using an antibody adsorption assay with S-HBs or L-HBs antigens. The purified antibodies induced in S-HBs vaccine- (blue) and L-HBs+Addavax- (red) immunized macaques were prepared at the 10-fold concentration of $IC_{50}$ values and mixed with S-HBs or L-HBs antigens at the indicated concentrations. HBV/NL-WT was incubated with antigen-adsorbed antibodies and used to infect G2/NT18-C cells. The infection efficiencies at the indicated concentrations of S-HBS or L-HBs antigens were assessed by measuring the NL activities of infected cells in 5 wells at 7 days after infection and indicated the mean ± SD at each concentration. The dose–response curves with the log (inhibitor) vs. normalized response model were drawn. Source data are provided as a Source Data file.

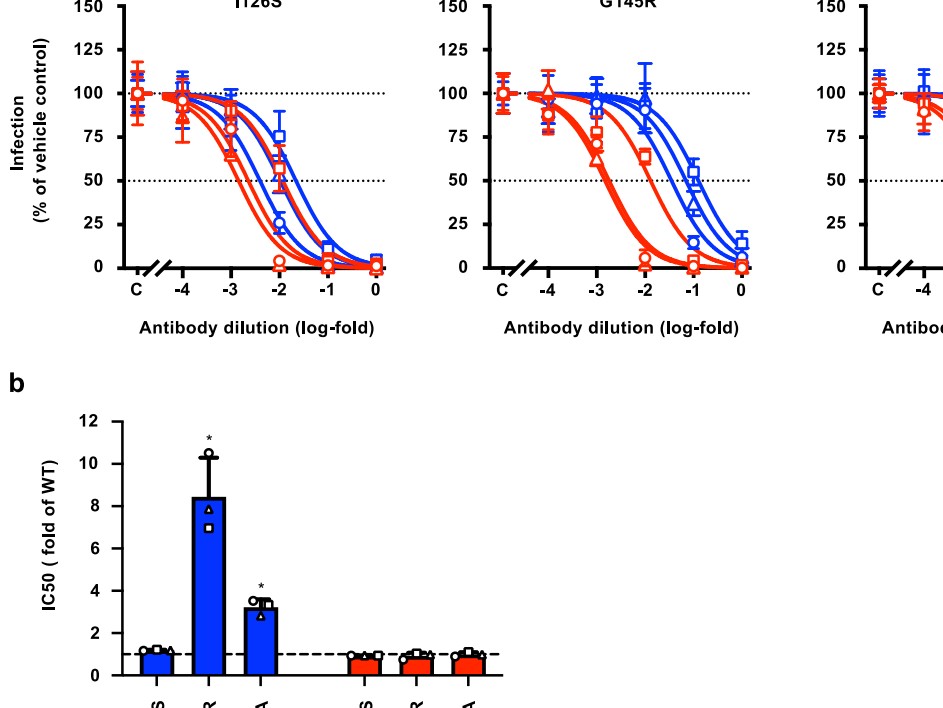

**Fig. 4 | Neutralizing activities against HBV with VEM. a** The neutralizing activities of induced antibodies to HBV with VEMs were evaluated by the HBV/NL infection system. The HBV/NL strains with VEMs, HBV/NL-I126S, -G145R, and -G145A (20 GEq/cell) were mixed with serially diluted antibodies from rhesus macaques immunized with S-HBs vaccine (blue) and L-HBs+Addavax (red) for 1 h at 37 °C and used to infect G2/NT18-C cells. The infection efficiencies were assessed by measuring the NL activities in infected cells at 7 days after infection. **b** The IC$_{50}$ values were calculated by drawing the dose–response curves with the log (inhibitor) vs. normalized response model and comparing the percentages to that of HBV/NL-WT. The mean ± SD of three animals is indicated. *$P = 0.0033$ by two-tailed paired $t$-test. Source data are provided as a Source Data file.

respectively (Fig. 5b and Supplementary Table 2). Regarding HBV/NL-GTD, the neutralizing activity of S-HBs vaccine-induced antibodies was lower than those of HBV/NL-GTA and -GTB (Fig. 5a, right panel), and the IC$_{50}$ value was 2.19 ± 0.489-fold that of HBV/NL-WT (GTC) (Fig. 5b and Supplementary Table 2). In experiments using L-HBs+Addavax-induced antibodies, the neutralizing activities against HBV/NL-GTA and -GTD were comparable to that against HBV/NL-WT (Fig. 5a). The neutralizing activity against HBV/NL-GTB was low, and the IC$_{50}$ value of antibodies was 7.18 ± 2.94-fold that of HBV/NL-WT (GTC) (Fig. 5b and Supplementary Table 2). These data indicate that the neutralizing effects of antibodies induced by the S-HBs vaccine and L-HBs+Addavax depend on HBV genotypes; attenuation of the neutralizing activities of S-HBs vaccine-induced and L-HBs+Addavax-induced antibodies was observed in the GTD and GTB strains, respectively.

**Region responsible for attenuated neutralizing effects on GTB**
To determine the region responsible for the attenuated neutralizing effects of L-HBs+Addavax-induced antibodies on the GTB strain, chimeric HBV/NL with various regions of the GTB strains was generated and used. The preS1 and preS2 regions of HBV/NL (GTC) were replaced with those of GTB, and the generated chimeric strains were named HBV/NL-pS1-GTB and -pS2-GTB, respectively (Fig. 6a). These HBV/NL strains were incubated with L-HBs+Addavax-induced antibodies, and the infection efficiencies were evaluated. The neutralizing activity to HBV/NL-pS1-GTB was attenuated similarly to that to HBV/NL-GTB, while the neutralizing activity to HBV/NL-pS2-GTB was almost identical to that to HBV/NL-WT (GTC), suggesting that the preS1 region is

responsible for the attenuated neutralizing effects of L-HBs+Addavax-induced antibodies on the GTB strain (Fig. 6b, d). Next, we divided preS1 into two regions; that including the RBD and that excluding the receptor-binding domain (exRBD), and generated HBV/NL strains containing these regions: HBV/NL-RBD-GTB and HBV/NL-exRBD-GTB, respectively (Fig. 6a). The neutralizing activity to HBV/NL-RBD-GTB was attenuated similarly and was close to that to HBV/NL-GTB or HBV/NL-pS1-GTB, whereas the neutralizing activity to HBV/NL-exRBD-GTB was similar to that to HBV/NL-WT (GTC). (Fig. 6c, d). These data suggest that five amino acid polymorphisms in the RBD specific for the GTB strain were responsible for the attenuated neutralizing effects of L-HBs+Addavax-induced antibodies (Supplementary Fig. 5). These data also indicate that the major antigenic region of L-HBs+Addavax-induced antibodies was in the RBD of the preS1 region. To identify the responsible amino acid, we replaced one of these amino acids with the GTC type and evaluated the neutralizing activities of L-HBs+Addavax-induced antibodies against these strains. The five strains of HBV/NL-RBD-GTB with GTC-type amino acids (-K35G, -E39N, -L45F, -H48N, and -N51H) showed attenuated neutralizing effects at a similar level to HBV/NL-RBD-GTB (Supplementary Fig. 7), and the responsible amino acid was not specified.

## Discussion
In this study, we established an HB vaccine that overcomes the shortcomings of the current HB vaccine. This vaccine induced the antibodies primarily to the preS1 region, which is not targeted by the S-HBs vaccine, in a nonhuman primate model. Using the infection

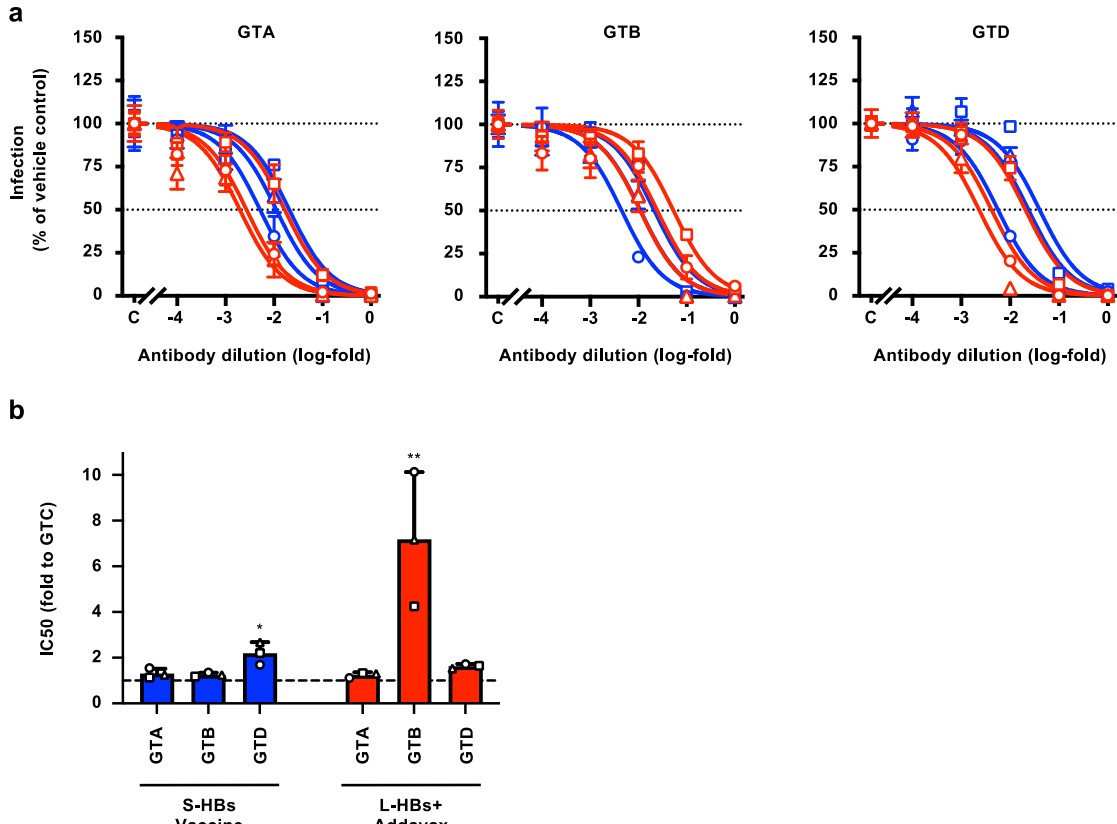

**Fig. 5 | Neutralizing activities against HBV strains of various genotypes. a** The neutralizing activities of induced antibodies to HBV strains of various genotypes were evaluated by the HBV/NL infection system. The HBV/NLs harboring the L-HBs region of GTA, GTB, and GTD were mixed with serially diluted antibodies from rhesus macaques immunized with S-HBs vaccine (blue) or L-HBs+Addavax (red) for 1 h at 37 °C and infected into G2/NT18-C cells. The infection efficiencies were assessed by measuring the NL activities in infected cells 7 days after infection. **b** The $IC_{50}$ values were calculated by drawing the dose−response curves with the log (inhibitor) vs. normalized response model and indicated the folds to that of HBV/NL-WT (GTC). The mean ± SD of three animals is indicated. *$P = 0.0139$ by two-tailed paired $t$-test. **$P = 0.0171$ by two-tailed paired $t$-test. Source data are provided as a Source Data file.

system of HBVcc and HBV reporter viruses, we demonstrated that HBV with VEMs escapes neutralization by antibodies induced by the S-HBs vaccine and proved that the HB vaccine comprising L-HBsAg could neutralize HBV infection with HBVs that contained VEMs to an extent that was similar to its effects on wild-type HBV. In addition, each vaccine compensates for the other's weaknesses in the attenuated neutralization of specific HBV genotypes. To the best of our knowledge, this is the first report to indicate the vaccine-escape capacity of VEMs with quantitative data.

Immunization of rhesus macaques with the HB vaccine comprising L-HBsAg+Addavax induced the anti-HBs antibodies without hepatotoxicity, and the antigenic region of induced antibodies was different from the S-HBs vaccine. Immunization with L-HBsAg+Addavax induced the antibodies mainly to the preS1 region, whereas immunization with the S-HBs vaccine induced the antibodies to S-HBs. These data suggest that the preS1 region exhibits more potent antigenicity than S-HBs when L-HBsAg is used as a vaccine. Immunization with L-HBsAg+K3-SPG also induced the antibodies to the preS1 region, but the induction level caused by this vaccine did not reach that by immunization with L-HBsAg+Addavax. Reflecting the antibody induction levels, the neutralizing effect against HBVcc by treatment with the serum samples of L-HBsAg+Addavax- and L-HBsAg+K3-SPG-immunized macaques were higher and lower than that of S-HBs vaccine-immunized macaques, respectively. K3-SPG is a K-type CpG oligodeoxynucleotide adjuvant encapsulated in schizophyllan. It is known to be a potent adjuvant for the induction of both humoral and cellular immune responses, especially for the induction of cytotoxic T lymphocytes[26,27]. Addavax, a formulation similar to MF59, is an oil-in-water emulsion adjuvant[28]. It is

used in seasonal influenza vaccines and is reported to enhance immune responses even in populations with a low response to vaccines, such as elderly people[29]. The characteristics of these adjuvants may be associated with differences in the efficiency of antibody induction.

For quantitative evaluation of the neutralizing activity, we used the HBV reporter virus infection system. The dose−response curves generated from the data on infection efficiencies indicated the potent neutralizing activity of the serum samples of L-HBsAg+Addavax-immunized macaques. The calculated $IC_{50}$ value of L-HBs+Addavax immunization was the lowest, followed by that of S-HBs vaccine immunization. Similar dose−response curves and $IC_{50}$ values were also obtained by treatment with antibodies purified from the serum samples of L-HBs+Addavax- and S-HBs vaccine-immunized macaques, suggesting that the antibodies induced by these HB vaccines were responsible for the observed neutralizing effects. The specificities for the neutralizing effects of purified antibodies were confirmed by the antibody adsorption assay. The neutralization of HBV infection by S-HBs vaccine-induced antibodies was incapacitated by adsorption with S-HBsAg, whereas neutralization by L-HBs+Addavax-induced antibodies was not affected. In contrast, neutralization by both S-HBs vaccine- and L-HBs+Addavax-induced antibodies was incapacitated by adsorption with L-HBsAg because this antigen comprises from preS1 to S-HBs regions. Taken together, these data suggest that the L-HBs +Addavax-induced antibodies recognize regions other than S-HBsAg and have potent neutralizing activities against HBV infection, similar to S-HBs vaccine-induced antibodies.

The amino acid polymorphisms at the specific position of the $a$-determinant in the S-HBs region are known to alter the binding affinity

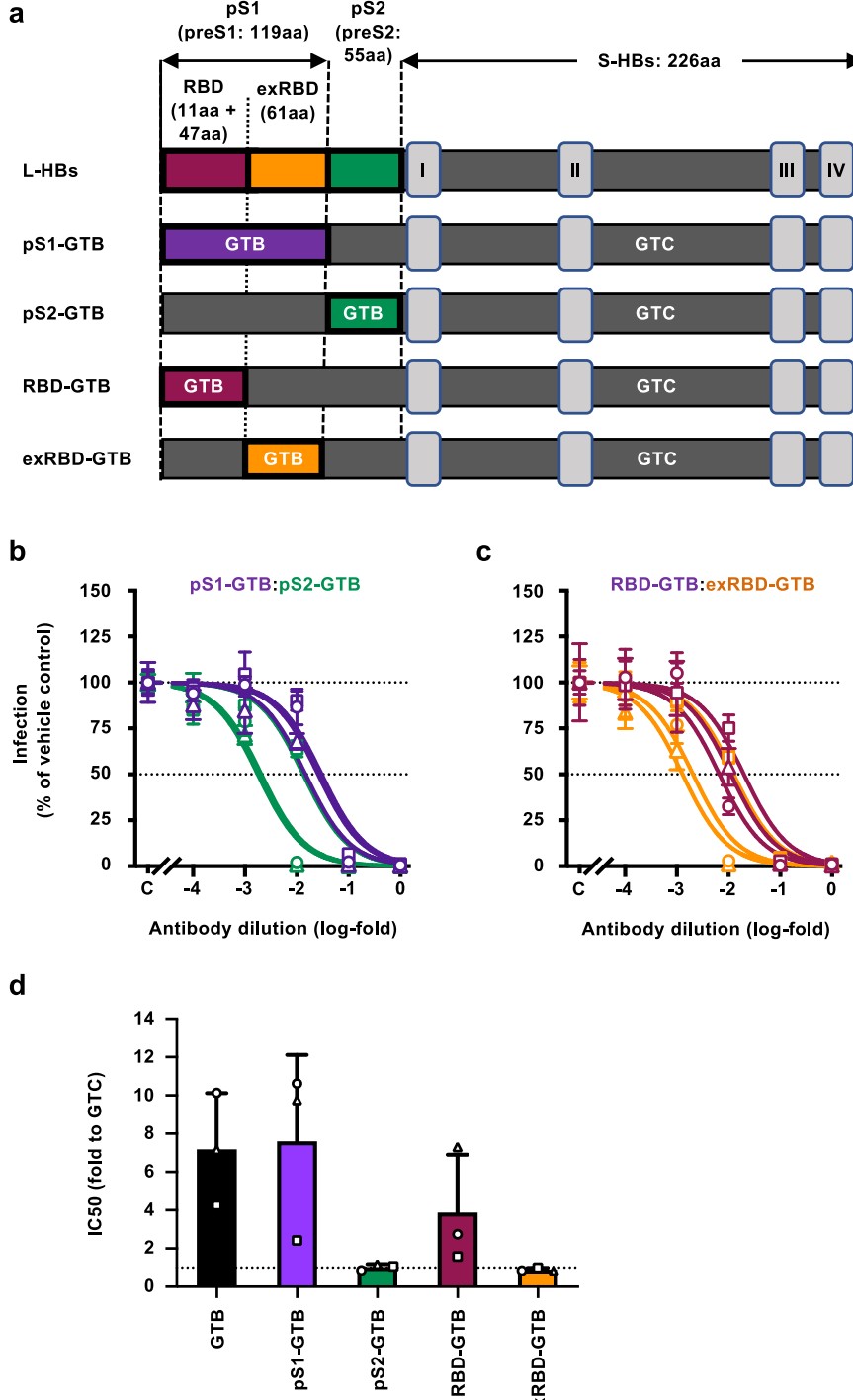

**Fig. 6 | Region responsible for attenuated neutralizing activities against GTB.**
**a** The schema of the structure of L-HBsAg and chimeric HBV/NL constructs. The number of amino acids included in regions of L-HBsAg is indicated, and transmembrane segments in S-HBsAg are labeled with roman numbers. **b** The neutralizing activities of L-HBs+Addavax-induced antibodies to HBV/NL with preS1 and preS2 of GTB were evaluated. The HBV/NLs harboring the preS1 (purple) or preS2 (green) region of GTB were mixed with serially diluted antibodies of rhesus macaques immunized with L-HBs+Addavax for 1 h at 37 °C and used to infect G2/NT18-C cells. The infection efficiencies were assessed by measuring the NL activities of infected cells in 5 wells 7 days after infection. **c** The neutralizing activities against

HBV/NL with the region including the receptor-binding domain (RBD) and excluding the receptor-binding domain (exRBD) of GTB were evaluated. The HBV/NLs harboring RBD (maroon) or exRBD (orange) of GTB were mixed with serially diluted antibodies for 1 h at 37 °C and were used to infect G2/NT18-C cells. The infection efficiencies were assessed by measuring the NL activities in infected cells 7 days after infection. **d** The IC50 values were calculated by drawing the dose–response curves with the log (inhibitor) vs. normalized response model and indicated the folds to that of HBV/NL-WT (GTC). The mean ± SD of three animals is indicated. Source data are provided as a Source Data file.

to the antibodies that recognize this region and neutralize HBV infection. These polymorphisms are designated VEMs, and the most common polymorphism associated with immune escape is G145R, which was detected in acute HB patients after the administration of the HB vaccine and immune globulin[9]. Since then, many other candidates for VEMs have been reported[13–15]. To assess the effects of VEMs on the neutralizing activities, we used the HBV/NL system with VEMs. In the HBV/NL system, amino acid substitutions in HBsAg can be introduced without altering the amino acids of the HB polymerase protein because these proteins are independently supplied from different plasmids. We evaluated the effects of the representative VEMs I126S, G145R, and G145A on neutralization by HB vaccine-induced antibodies. The neutralizing activity of S-HBs vaccine-induced antibodies was substantially decreased by the introduction of G145R into HBV; the $IC_{50}$ value was increased to approximately 8.5-fold that of HBV/NL-WT. Similarly, G145A also increased the $IC_{50}$ values, although attenuation of the neutralizing activity to HBV with I126S was minimal in this assay. These data indicate that S-HBs vaccine-induced antibodies neutralize HBV with G145R or G145A when the antibody titer is high enough but cannot neutralize them when the antibody titer is low. In contrast, in the case of L-HBs+Addavax-induced antibodies, the $IC_{50}$ values for HBV/NL with these VEMs were not affected. These data suggested that the neutralizing antibodies induced by L-HBs+Addavax can prevent HBV infection with VEMs that are not neutralized sufficiently by antibodies induced by the currently available S-HBs vaccine. The attenuation of neutralizing activity by S-HBs vaccine-induced antibodies against HBV with G145R was also confirmed in S-HBs vaccine-immunized human individuals. Therefore, the HB vaccine consisting of L-HBsAg+Addavax was proven to compensate for the shortcomings of the currently available HB vaccine.

HBV is classified into at least eight genotypes, GTA to GTH, based on its genetic diversity, and differences in clinical features and outcomes have been reported among these genotypes[30,31]. Among them, the HBV strains GTA, GTB, GTC, and GTD are predominant worldwide. Regarding S-HBs vaccine-induced antibodies, genotype-dependent attenuation of neutralizing activity has been reported[32]. The required titer of HB vaccine-induced antibodies was higher when neutralizing HBV of the different genotypes from the vaccinated one than when neutralizing the same genotype. In the present study, to quantify the genotype dependency of the neutralizing activity, we used a chimeric reporter virus whose envelope proteins were replaced with those of the GTA, GTB, or GTD strain. Regarding neutralization by S-HBs vaccine-induced antibodies, the $IC_{50}$ values for GTA and GTB were comparable to that for GTC (the genotype used in the vaccine). The $IC_{50}$ value for GTD was more than 2-fold that for GTC. In the case of L-HBs +Addavax-induced antibodies, attenuated neutralization was observed for GTB, and the $IC_{50}$ value increased to approximately 7-fold that for GTC. These data suggested that the antibodies induced by a single HB vaccine did not neutralize HBV infections of all genotypes at the same level and that genotypes with weak neutralizing activity differed from each other. If the level of antibody induction is not high or if the titer of the induced antibodies decreases over time, neutralization of the HBV genotypes other than those used in the vaccination may not be feasible. The various chimeric HBV reporter viruses between GTB and GTC revealed that amino acid polymorphisms of the RBD in the preS1 region are responsible for the attenuated neutralizing activity of L-HBs +Addavax-induced antibodies, although we could not specify the responsible amino acid in this region. These data also suggest that this region contains the major antigenic region of antibodies induced by L-HBs+Addavax and is distinct from the antigenic region of the S-HBs vaccine, the *a*-determinant in the S-HBs region. Therefore, simultaneous or sequential immunization of these HB vaccines may help to induce antibodies that can neutralize multiple genotypes of HBV.

In conclusion, using L-HBsAg, we established an HB vaccine that induces the antibodies capable of neutralizing HBV with VEMs that

cannot prevent infection by antibodies induced by the currently available S-HBs vaccine. In addition, the HBV genotypes that exhibited attenuated neutralization by induced antibodies differed between these vaccines. Thus, the combined use of these HB vaccines may be able to induce antibodies that neutralize HBV variants, including strains with VEMs or strains of multiple HBV genotypes. HBV strains of other genotypes or with future emerging VEMs will be neutralized by antibodies induced by either of these vaccines. The combination of these vaccines that includes different adjuvants may be effective in inducing the production of antibodies even in a population that has low or no humoral response to the S-HBs vaccine as observed in approximately 10% of vaccinated adults. Further investigation will be needed to establish the protocol for the combined use of these vaccines to induce the efficient production of neutralizing antibodies. In addition, the infection systems of HBV reporter viruses used in this study were useful to evaluate the neutralizing activity of HB vaccine-induced antibodies accurately and quantitatively, and this evaluation will be needed to discuss the potency and characteristics of the HB vaccines and VEM- or genotype-dependent attenuation of the neutralizing activity of induced antibodies.

## Methods

### HB vaccines and adjuvants

The commercially available S-HBs vaccine, Bimmugen, was purchased from KM Biologics (Kumamoto, Japan). This vaccine uses the yeast-derived S-HBsAg of the HBV GTC strain and contains aluminum hydroxide as an adjuvant. The yeast-derived L-HBsAg (BCL-AG-001) was obtained from Beacle, Inc. (Kyoto, Japan)[16,18]. The K-type CpG oligodeoxynucleotide adjuvant wrapped by schizophyllan (K3-SPG) was prepared as previously described[26,27]. A squalene-based oil-in-water nanoemulsion adjuvant (Addavax) was purchased from Invivogen (San Diego, CA). This adjuvant is known to have a formulation similar to that of MF59.

### Monkey experiments

In this study, nine rhesus macaques (3–5 years of age, male or female, and approximately 5 kg body weight) were used as a nonhuman primate model for immunization with HB vaccines. All monkeys were supplied from colonies at the Center for the Evolutionary Origins of Human Behavior, Kyoto University. The monkeys were reared in outdoor group cages with wooded toys provided as environmental enrichment. They were fed apples, potatoes, and commercial monkey diets. They were able to access water ad libitum. They had health records from birth with yearly health checkups. We confirmed that they were free of B virus, simian immunodeficiency virus, simian T-cell leukemia virus, and simian retrovirus. The animal experiments were conducted using protocols and experimental procedures that were approved by the Animal Welfare and Animal Care Committee of Kyoto University (approval number: 2017-158, 2018-122, 2019-144, 2020-105, 2021-159) and were carried out in accordance with the Guidelines for Care and Use of Nonhuman Primates (Version 3) by the Animal Welfare and Animal Care Committee of Kyoto University. This guideline was prepared based on the provisions of the Guidelines for Proper Conduct of Animal Experiments (June 1, 2006; Science Council of Japan) as well as Fundamental Guidelines for Proper Conduct of Animal Experiment and Related Activities in Academic Research Institutions [Notice No. 71 of the Ministry of Education, Culture, Sports, Science and Technology dated June 1, 2006], in accordance with the recommendations of the Weatherall report: The use of nonhuman primates in research (http://www.acmedsci.ac.uk/more/news/the-use-of-nonhuman-primates-in-research/).

HB vaccines were introduced subcutaneously to immunize rhesus macaques under ketamine anesthesia with medetomidine, and then its antagonist atipamezole was administered at the end of the procedure. Blood samples were collected under anesthesia as described above. To

assess the potential immediate side effects caused by the vaccines, veterinary observation, including any clinical and hematological abnormalities as well as the presence of skin abnormalities such as flare, rash, or edema, was carefully performed.

## Cell culture

HepG2 cells were obtained from the European Collection of Authenticated Cell Cultures (catalog number; EC85011430, ECACC, Salisbury, UK) and cultured in MEM supplemented with 10% fetal calf serum[22]. Sodium taurocholate cotransporting polypeptide (NTCP)-transduced HepG2 cells; G2/NT18-C and HepG2-NTCPsec+, were used as described previously[24,25].

## Enzyme-linked immunosorbent assay

The titer of anti-S-HBs antibodies was measured using an Enzygnost Anti-HBs II (Siemens Healthcare Diagnostics, Tokyo, Japan). To measure the titers of anti-preS1 and anti-L-HBsAg, laboratory-made enzyme-linked immunosorbent assays were used[33]. Briefly, 1 μg/mL of preS1-peptide (N-terminal 12–119 aa of preS1 region, BCL-AGS1-02, Beacle) or 1 μg/mL of L-HBsAg (Beacle) in phosphate-buffered saline (PBS) containing 0.02% $NaN_3$ (PBSN) was adsorbed onto MaxiSorp 96-well plates (Nunc Immunoplate F96 Cert MaxiSorp, Thermo Fisher Scientific, Nunc A/S, Roskilde, Denmark) during overnight incubation at 4 °C. Wells were washed three times with wash buffer (PBS containing 0.05% Tween 20), blocked with 0.2% BSA-supplemented PBSN, and incubated for more than 24 h. The plates were washed three times prior to use. Serum diluted with 0.2% BSA-supplemented PBST containing 0.02% $NaN_3$ (PBSTN) was added to each well, and the plates were incubated overnight at 4 °C. The supernatant was then discarded, and each well was washed three times. HRP-conjugated anti-human IgG (P0214, Dako Cytomation, Glostrup, Denmark) was added at the concentration of 0.2 μg/mL and incubated for 2 h at 37 °C prior to the addition of substrate (TMB Substrate Kit, Thermo Scientific, IL). Thirty minutes later, color development was stopped by the addition of 2 N $H_2SO_4$ at 100 μL/well, and then the optical density was measured at 450 nm in a microplate reader (SUNRISE, Tecan, Grödig, Austria).

## Production and infection of cell culture-generated HBV

The plasmid for HBVcc was generated with the HBV GTC strain (accession number: AB246345)[22]. This plasmid encodes a replication-competent HBV molecular clone with a 1.38-fold genome length (Supplementary Fig. 2a). The HBV clone was transfected into HepG2 cells by using Lipofectamine 3000 Reagent (Thermo Fisher Scientific, Waltham, MA), and the culture medium was harvested 1 week after transfection. The collected medium was passed through a 0.45-μm filter to remove cell debris and purified by an iodixanol density gradient. The peak fraction of infectivity in the gradient was used as an inoculum. The HBV DNA titer of the inoculum was measured by real-time PCR targeting the HBs region after treatment with DNase (RQ1 RNase-Free DNase, Promega, Madison, WI).

The prepared viruses were mixed with rhesus macaque serum, incubated at 37 °C for 1 h, and inoculated onto HepG2-NTCP-sec+ cells in the presence of 4% PEG8000 and 2% dimethyl sulfoxide for 16 h. Twelve days after infection, infected cells were treated with rabbit polyclonal anti-HBc IgG fraction (catalog number; HBP-023-9, Austral Biologicals, San Ramon, CA) at the concentration of 1.0 μg/mL followed by staining with Alexa Fluor 555-conjugated anti-rabbit IgG (catalog number; A32732, Thermo Fisher Scientific) at the concentration of 1.0 μg/mL after fixation and permeabilization. Nuclei were also stained with 4′,6-diamidino-2-phenylindole. The infected cells were visualized and captured with a BZ-X710 fluorescence microscope (Keyence, Osaka, Japan). To estimate the infection efficiencies, the area size of staining was quantified with the built-in software BZ-3HA ver. 1.3.0.3 (Keyence).

## Recombinant HBV reporter virus system

Regarding the recombinant HBV reporter virus (HBV/NL) infection system, HBV-NL plasmids encoding the 1.2-fold HBV genome replacing the HBe/HBc region with NL and HBV-dEdelS plasmids encoding the 1.2-fold HBV genome lacking the encapsidation signal and the start codons of ORF of all HBsAg species were prepared (Supplementary Fig. 2b)[22,34]. We used the GTC clone (accession number: AB246345) for the generation of these plasmids. One week after the transfection of these plasmids into HepG2 cells, the HBV/NL in the culture medium was harvested. The collected medium was passed through a 0.45-μm filter to remove cell debris and purified by an iodixanol density gradient. The peak fraction of infectivity in the gradient was used as an inoculum, and the same inoculum was used to assess neutralization by serum or purified antibodies induced by all vaccines. The viral titer was measured by real-time PCR with a primer and probe set targeting the NL region after treatment with DNase (RQ1 RNase-Free DNase). One week after infection of HBV/NL into G2/NT18-C cells, the infection efficiencies were estimated by measuring the luciferase activities in infected cells using a Nano-Glo Luciferase Assay System (Promega) after lysis of cells with Passive Lysis Buffer (Promega).

The HBV/NL strain with VEMs was also generated. In this system, all HBsAg species were provided from the HBV-NL plasmid. Thus, the amino acid mutations I126S, G145R, and G145A were introduced into the HBV-NL plasmid by site-directed PCR, and the fragment was swapped by digestion with the restriction enzymes BstEII and BstBI (Supplementary Fig. 2b). The chimeric HBV-NL plasmid encoding the L-HBs region of GTA, GTB, and GTD was also generated by replacing the L-HBs region with those of the GTA (accession number: LC488828), GTB (accession number: AB246341), and GTD strains by digestion with the restriction enzymes BstEII and BstBI (Supplementary Fig. 2b). The sequence of the L-HBs region of the GTD strain is identical to that of HBV derived from HepG2.2.15 cells[35]. The chimeric HBV-NL plasmids HBV/NL-pS1-GTB, -pS2-GTB, -RBD-GTB, and -exRBD-GTB were also generated by site-directed PCR and fragment swapping. The plasmids for HBV/NL-RBD-GTB with GTC-type amino acids were generated by site-directed PCR. The alignment of the amino acid sequences of the L-HBs region is provided in Supplementary Fig. 5.

## Purification of antibodies and antibody adsorption assay

The induced antibodies in serum samples of rhesus macaques were purified by Protein G HP SpinTrap (Cytiva, Tokyo, Japan) following the manufacturer's instructions. To estimate the amount of purified antibodies, the protein concentrations of the purified antibodies were quantified by a Pierce BCA Protein Assay Kit (Thermo Fisher Scientific). For the antibody adsorption assay, S-HBsAg (recombinant HBsAg adr produced in CHO cells, ProSpec, Rehovot, Israel) and L-HBsAg (Beacle) were used.

## Human serum samples

Serum samples were obtained with written informed consent without compensation from three individuals (37–42 years old, 1 male and 2 females) who were vaccinated with the commercially available S-HBs vaccine in Japan, Bimmugen. The vaccine-induced antibodies were purified from serum samples and evaluated for the neutralization of HBV reporter viruses. This experiment was approved by the Ethics Committees of the institute (approval number 780 from the National Institute of Infectious Diseases).

## Statistical analysis

Statistical analysis was performed by a ratio paired t-test using GraphPad PRISM 8 software (GraphPad Software, La Jolla, CA). The $IC_{50}$ values were calculated by generating the dose–response curves with the log (inhibitor) vs. the normalized response model. Results with P values of <0.05 were considered statistically significant.

**Reporting summary**

Further information on research design is available in the Nature Research Reporting Summary linked to this article.

## Data availability

Source Data are provided with this paper.

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

## Acknowledgements

This work was supported by grants for Research Programs on Hepatitis (JP19fk0310120 [T. Kato, H.A., R.S., Y.T., H.Y.], JP21fk0310103 [T. Kato], JP22fk0310503 [T. Kato, HA], and JP22fk0310517 [T. Kato]) from the Japan Agency for Medical Research and Development, AMED, and by the Taiju Life Social Welfare Foundation [T. Kato]. The funders had no role in the study design, data collection or interpretation, or the decision to submit the work for publication. The authors wish to thank Dr Marc Peter Windisch (Institut Pasteur Korea, Seoul, South Korea) for providing the HepG2-NTCPsec+ cell line, and Mses Hitomi Igarashi and Kaoru Tsuji for their technical assistance.

## Author contributions

H.A. and T. Kato conceived this study. A.W., A.M., M.M., T. Kiyohara, K.Y., N.Y., H.H.A., R.S., H.A., and T. Kato carried out the experiments. A.W., A.M., H.A., and T. Kato discussed and interpreted the results. A.W., A.M., H.A., and T. Kato wrote the manuscript. T.T., K.M., H.N., K.S., Y.G., and K.J.I. provided the key materials and instructions for use. H.Y., M.M., K.I., and Y.T. supervised the experiment and project.

## Competing interests

Y.G. is an employee of Beacle, Inc. The other authors declare no competing interests.
