## [Peer review file · Nature Communications]

REVIEWER COMMENTS

Reviewer #1 (Remarks to the Author):

This study by Washizaki et al. is evaluating the immunogenicity and neutralization potential of a new HBV vaccine. Current HBV vaccines employ small HBsAg. Additional antigens, including the receptor binding domain of HBsAg, could improve protection from variants of HBV. The authors find that large HBsAg (PreS1-PreS2-S) induces unique antibodies that primarily target the PreS1 receptor binding domain of HBsAg. In addition, the authors use a very nice reporter virus system to test different HBsAg sequences and the ability of the induced antibodies to neutralize these viruses. The data presented is promising, but there are a few significant flaws in the study design that need to be corrected. In addition, the overall impact of the findings are still somewhat unclear and more concrete information on the impact of HBV VEMs to global health is needed.

Comments:

- 1) It remains unclear how readily HBV with VEMs actually evade conventional vaccine-induced antibody responses. In fact, the authors state that no quantitative measure of this has been conducted and then show that although IC50s increase, these VEMs are definitely still readily neutralized by conventional vaccine responses. What is the global burden of VEM transmission? Outside of the phenomenon of vaccine non-responders, it is generally accepted that neutralization/protection from HBV is nearly complete in responding individuals.
- 2) There is some missing safety data. Serum ALT/AST/ALKP levels should be shown following administration of L-HBs vaccine.
- 3) One of the reasons safety data should be included is that inclusion of the Pre-S1 region in your vaccine means most of your protein finding the blood stream will bind the liver. Since Hepcludex is fairly safe, it is unlikely that safety of your vaccine is an issue, but this absolutely should be addressed with quantitative data. Does Hepcludex administration lead to neutralizing antibodies against Pre-S1 RBD? If so, this seems like a much easier path forward since this is already approved in EU for the treatment of HDV.
- 4) Route of vaccine administration should be clearly outlined in the main text and not left only in the methods. This is an important aspect of the study.
- 5) Fig 1B: How do the authors explain the induction of anti-L-HBs antibodies in rhesus macaques immunized with S-HBs vaccine, since it only includes the S ORF and not PreS1 or PreS2? This doesn't seem to make sense.
- 6) Fig 2: Authors need to be explicit here that the same HBVcc collection was used across each experiment, since each HBVcc collection will have different (all high) levels of HBsAg containing subviral particles that will affect neutralization results.

- 7) Fig 2: Figure is very small and difficult to interpret. Microscopy should be moved to supplemental materials and representative data from all vaccinated animals should be shown. What was the variability between animals within groups?
- 8) Figs 2 and 3A: Serum can have non-specific inhibitory effects on infection assays. Unfortunately, the authors do not provide control sera in their experiments that are from HBsAg naive animals. This is a missing and crucial control in their experiments.
- 9) Figs 3-6: Authors miss a critical opportunity here to normalize for antibody (protein) concentration to determine if neutralization potential within the L-HBs population of antibodies exists between macaques. Why are purified antibodies (adsorbed or non-adsorbed) not quantified and normalized?
- 10) HBV RBD is <60 amino acids. Authors should expand their study to include: 1) AA alignment between the genotype reference sequences used in Figs 5 and 6, and 2) expanded experiments defining the AA(s) responsible for escape from L-HBs-specific antibodies.
- 11) Since a simple HBV genotype change can abrogate the neutralization potential of L-HBs-specific antibodies, it seems likely that different genotypes of the virus will simply evolve new "VEMs" that are simply the GTB sequence in Pre-S1. Clearly that virus has no fitness disadvantage. Overall, won't it be easier for HBV to evolve to evade your vaccine compared to the contemporary vaccine?

Reviewer #2 (Remarks to the Author):

Background: The intent of treatment for HBV is usually limited to suppressing viral replication, as complete elimination of the virus can rarely be achieved with current therapies. Therefore, preventative vaccination is considered the best strategy for managing HBV infection. However, current vaccines targeting the S-HBs antigen fail to generate strong humoral responses in 10% of vaccinated individuals, and a small number of vaccine-escape mutations (VEMs) have been reported that evade neutralization.

Noteworthy results: Washizaki et al. developed a complementary vaccine that targets the large HBs antigen, which contains the NTCP binding site. They tested the vaccine in rhesus macaques and in cell culture and reported neutralization activity similar to that of the current HBV vaccine but unaffected by current VEMs.

Significance: The authors do not propose the vaccine as a replacement for the current widely used HBV vaccine but rather as a complementary vaccine intended to overcome various shortcomings and gaps in the protection offered by the standard vaccine. Part of this strategy is to include the NTCP-binding region in the vaccine target, which might improve the efficacy of the vaccine by directly targeting a key functional region of the S protein. The L-HBsAg region is only present on Dane particles, whereas S-HBsAg is also present on the vastly more abundant sub-viral particles, which might serve as a form of immune decoy or sink. However, the current vaccine requires three doses, and the regimen described by the authors also appears to require three doses, so it is unclear whether a six-dose vaccine schedule against essentially the same target is sufficiently practical or necessary. A related approach that achieves

many of the same objectives is to target the NTCP receptor instead (e.g., PMID: 34363922), which directly blocks infection by preventing the virus from entering the cell and is less affected by VEMs or differences among viral genotypes. Washizaki et al. examined the most common genotypes (A, B, C, and D), which is more than sufficient to support their conclusions, but the study might have greater significance if the same results could be confirmed for some of the less common genotypes and minor strains, especially if the vaccine provides notably better protection than the standard vaccine. Stable HBVcc cell lines producing high titers of HBV genotypes A2, B2, C1, E, F1b and H are available (PMID: 34985994).

- Does the work support the conclusions and claims, or is additional evidence needed?

The authors' conclusions appear to be sound with sufficient supporting evidence.

- Are there any flaws in the data analysis, interpretation and conclusions

The data analysis does not appear to contain flaws. However, the conclusions feel incomplete without examination of other genotypes and evidence of long-term protection. It is also unclear how and to whom the vaccine would be administered in relation to the standard vaccine. The authors note that the response to the vaccine was carefully monitored in the macaques, but prediction of potential off-target effects in humans should also be considered.

- Is the methodology sound? Does the work meet the expected standards in your field?

The methodology appears to be sound and to meet high standards.

- Is there enough detail provided in the methods for the work to be reproduced?

It should be possible to reproduce the work based on the level of detail provided.

Reviewer #3 (Remarks to the Author):

In this manuscript, Kato et al studied a L-HBsAg-derived vaccine candidate and its efficacy against previously described vaccine escape mutants (VEM) and various HBV genotypes in comparison to standard S-HBsAg vaccine. They showed that the L-HBsAg induced antibody response is more potent against VEM by directing the antibody response mostly to the pre-S1 region instead of the "a" domain of the HBsAg, where the VE mutations reside. On the other hand, it appears to be more variable and less effective against HBV genotypes other than the genotype of the vaccine. This is not surprising given the high level of variations in the pre-S1 region. Previous studies have attempted to generate L-HBsAg-based vaccine on account of the same rationale, but the yield of production and quality of the antigen have

been marginal and thus never gain much traction. This study is useful in advancing the goal of developing this type of vaccine, which may have an intrinsic advantage as discussed by the authors. The authors did a nice job in characterizing the induced antibodies in both rhesus and human and clearly defined the different antibody responses between the L-HBsAg and the standard S-HBsAg vaccines. The data are reasonably convincing and the analysis and interpretation are appropriate. Based on this study, it is clear that the L-HBsAg vaccine is more effective against VEMs but less so against other HBV genotypes, which is more of a problem if the aim is to produce a widely effective vaccine. VEMs have not been a serious problem and most of HBV vaccine induced antibodies are sufficiently potent to neutralize many of the VEMs. Thus I am not sure whether this vaccine candidate is any better than what we have. A few other major comments about this study.

1. It is probably more appropriate to compare the standard S-HBs vaccine together with the same adjuvant (K3-SPG or Addavax) against the L-HBsAg + adjuvant. It is clear the adjuvant can significantly enhance the antibody response.
2. Would administration of both S-HBs and L-HBsAg vaccines generate an additive antibody response? It would be of interest to study such a combination.
3. The recent approval of the Dynavax HBV vaccine (HEPLISAV-B) is formulated with an adjuvant that is a TLR agonist and can induce much higher tiers of anti-HBs, and requires only two doses of vaccine. It would be worthwhile to compare the L-HBsAg based vaccine to this one.

Minor points:

1. In the section on the HBV genotypes, the authors equate neutralizing activity was higher with higher fold of IC50. But the fact is the lower the IC50 the more potent the neutralizing activity. The authors should be careful about the description.
2. In Suppl Table S1 and S2, the last column is labeled as % of WT, but the number given is a fraction, which should be revised.
3. The Discussion repeats much of Results and should be more concise and focus on the importance and relevance of the data in the context of the current knowledge. The authors provided a less than adequate and unbiased discussion of their study.

Response to reviewer's comments

Reviewer #1 (Remarks to the Author):

This study by Washizaki et al. is evaluating the immunogenicity and neutralization potential of a new HBV vaccine. Current HBV vaccines employ small HBsAg. Additional antigens, including the receptor binding domain of HBsAg, could improve protection from variants of HBV. The authors find that large HBsAg (PreS1-PreS2-S) induces unique antibodies that primarily target the PreS1 receptor binding domain of HBsAg. In addition, the authors use a very nice reporter virus system to test different HBsAg sequences and the ability of the induced antibodies to neutralize these viruses. The data presented is promising, but there are a few significant flaws in the study design that need to be corrected. In addition, the overall impact of the findings are still somewhat unclear and more concrete information on the impact of HBV VEMs to global health is needed.

Comments:

1) It remains unclear how readily HBV with VEMs actually evade conventional vaccine-induced antibody responses. In fact, the authors state that no quantitative measure of this has been conducted and then show that although IC₅₀s increase, these VEMs are definitely still readily neutralized by conventional vaccine responses. What is the global burden of VEM transmission? Outside of the phenomenon of vaccine non-responders, it is generally accepted that neutralization/protection from HBV is nearly complete in responding individuals.

Response:

In this study, by using the precise and quantitative HBV reporter virus infection system, we found that the IC₅₀ value of antibodies induced by the currently available S-HBs vaccine against HBV with VEMs was approximately 3 to 8 times higher than that against wild-type HBV. These data indicate that such antibodies neutralize HBV with VEMs when the antibody titer is high enough but cannot neutralize them when the antibody titer is low. These data also reasonably explain the emergence of HBV with VEMs among the population vaccinated with the S-HBs vaccines described in references 12, 13, and 14. In addition, this is the first report of quantified data on the vaccine escape capacity of VEM, and these data will be useful for further discussion of the vaccine escape capacity of other mutations.

To emphasize this point, we added the following sentence to the Discussion: 'These data indicate that S-HBs vaccine-induced antibodies neutralize HBV with G145R or G145A when the antibody titer is high enough

but cannot neutralize them when the antibody titer is low.' (page 22, lines 389-391).

2) *There is some missing safety data. Serum ALT/AST/ALKP levels should be shown following administration of L-HBs vaccine.*

Response:

Following the reviewer's suggestion, we included the safety data of the vaccines. During immunization with vaccines, the liver enzyme levels, including the levels of alanine aminotransferase (ALT), aspartate aminotransferase (AST), and gamma-glutamyl transpeptidase (GGT), were monitored. We could not measure ALKP levels in animal samples. Thus, instead, we measured GGT levels. Vaccine-associated abnormalities were not observed in immunized animals. The obtained data are indicated in Supplementary Figure 1 and explained in the Results as follows: **'During immunization with vaccines, vaccine-associated clinical abnormalities or severe increases in liver enzyme levels, including levels of alanine aminotransferase (ALT), aspartate aminotransferase (AST), and gamma-glutamyl transpeptidase (GGT), were not observed (Supplementary Fig. 1).'** (Page 10, lines 165-169).

3) *One of the reasons safety data should be included is that inclusion of the Pre-S1 region in your vaccine means most of your protein finding the blood stream will bind the liver. Since Hepcludex is fairly safe, it is unlikely that safety of your vaccine is an issue, but this absolutely should be addressed with quantitative data. Does Hepcludex administration lead to neutralizing antibodies against Pre-S1 RBD? If so, this seems like a much easier path forward since this is already approved in EU for the treatment of HDV.*

Response:

Regarding Hepcludex, we have no data. We also have no data regarding the distribution of the L-HBs vaccine in the body after immunization. However, since it was introduced subcutaneously, it is expected to be maintained in the immunized area. In addition, as shown above, we did not observe any vaccine-associated abnormalities in liver enzyme levels in the vaccinated animals.

4) *Route of vaccine administration should be clearly outlined in the main text and not left only in the methods. This is an important aspect of the study.*

Response:

Following the reviewer's suggestion, we described the administration route of vaccines in the Results as follows: 'These vaccines were administered 3 times subcutaneously at 4 and 20 weeks after the initial vaccination following the protocol of the 3-dose HB vaccine series in humans, and the observation was continued until 26 weeks after the initial vaccination (Fig. 1).' (page 10, line 163) and in Figure Legend of Figure 1 as follows: 'These vaccines were administered 3 times subcutaneously at 0, 4, and 20 weeks, and the observation was continued until 26 weeks after the initial vaccination.' (page 35, line 604).

5) *Fig 1B: How do the authors explain the induction of anti-L-HBs antibodies in rhesus macaques immunized with S-HBs vaccine, since it only includes the S ORF and not PreS1 or PreS2? This doesn't seem to make sense.*

Response:

As explained in the Introduction, L-HBsAg includes the regions from preS1 to S-HBs. Therefore, the antibodies induced by the S-HBs vaccine bind to L-HBsAg (Figure 1b) but do not bind to preS1 (Figure 1c). For the same reason, immunization with the L-HBs vaccine induces the production of antibodies against S-HBs (Figure 1a).

6) *Fig 2: Authors need to be explicit here that the same HBVcc collection was used across each experiment, since each HBVcc collection will have different (all high) levels of HBsAg containing subviral particles that will affect neutralization results.*

Response:

As mentioned by the reviewer, we used the same inoculum prepared by the density gradient to assess neutralization by serum or purified antibodies induced by all vaccines. We explained this in the Methods section as follows: 'The peak fraction of infectivity in the gradient was used as an inoculum, and the same inoculum was used to assess neutralization by serum or purified antibodies induced by all vaccines.' (page 32, lines 550-551).

7) Fig 2: Figure is very small and difficult to interpret. Microscopy should be moved to supplemental materials and representative data from all vaccinated animals should be shown. What was the variability between animals within groups?

Response:

The data of all vaccinated macaques (3 animals in each vaccination) are indicated in Figure 2a. To address the reviewer's comment, we provided the quantified data of the stained area with anti-HBc antibodies in Figure 2b, indicating the variability between animals. These data were explained in the Results as follows: **'The number of HBc-positive cells was higher by viral infection after treatment with serum samples from L-HBs+K3-SPG-immunized macaques and was lower by viral infection after treatment with serum samples from L-HBs+Addavax-immunized macaques than with those from S-HBs vaccine-immunized macaques (Fig. 2b).'** (Page 12, lines 197-201).

8) Figs 2 and 3A: Serum can have non-specific inhibitory effects on infection assays. Unfortunately, the authors do not provide control sera in their experiments that are from HBsAg naive animals. This is a missing and crucial control in their experiments.

Response:

We agree that serum sometimes shows an antibody-independent neutralizing effect. To address this comment, we assessed the neutralizing effect of preimmune serum samples obtained from the enrolled macaques. Treatment with preimmune serum showed no inhibitory effects on HBV/NL-WT infection. The data of this experiment are provided in Supplementary Fig. 3 and explained as follows: **'The preimmune serum samples of these animals indicated no inhibitory effect of HBV/NL-WT infection (Supplementary Fig. 3).'** (page 13, lines 217-218).

9) Figs 3-6: Authors miss a critical opportunity here to normalize for antibody (protein) concentration to determine if neutralization potential within the L-HBs population of antibodies exists between macaques. Why are purified antibodies (adsorbed or non-adsorbed) not quantified and normalized?

Response:

Following the reviewer's comment, we measured the protein concentrations of antibodies purified from the endpoint serum samples of immunized macaques. The protein concentrations of antibodies purified from S-HBs vaccine- and L-HBVs+Addavax-immunized macaques were comparable, while those purified from L-HBVs+K3-SPG-immunized macaques were slightly lower than those. These data seem to be consistent with the total amount of induced antibodies indicated in Figure 1 and may reflect the lower induction of antibodies by immunization with L-HBVs+K3-SPG. However, we did not use these data to normalize for inhibitory effects because all antibodies, vaccine-induced and those not vaccine-induced, were included in these values, and the overestimation of induced antibody levels could not be excluded. The data of this experiment are provided in Supplementary Figure 4 and are explained as follows: **'The protein concentrations of antibodies purified from S-HBs vaccine- and L-HBVs+Addavax-immunized macaques were comparable, while that purified from L-HBVs+K3-SPG-immunized macaques was slightly lower than those (Supplementary Fig. 4).'** (page 13, lines 222-225).

10) HBV RBD is <60 amino acids. Authors should expand their study to include: 1) AA alignment between the genotype reference sequences used in Figs 5 and 6, and 2) expanded experiments defining the AA(s) responsible for escape from L-HBs-specific antibodies.

Response:

The alignment of amino acids in the preS1-HBs regions of HBV strains used in this study is provided in Supplementary Fig. 5. We found five GTB-specific amino acids in the RBD: G35K, N39E, F45 L, N48H, and H51N. To identify the amino acid responsible for the reduced inhibitory effect against the GTB strain, one of these amino acids in HBV/NL-RBD-GTB was replaced with the GTC type, and the neutralizing activities of L-HBs+Addavax-induced antibodies were assessed. We found that all assessed replacements of these amino acids with the GTC-type in HBV/NL-RBD-GTB indicated attenuated neutralizing effects at a similar level to HBV/NL-RBD-GTB. Therefore, we could not specify the amino acid responsible. Since these amino acids are located adjacent to each other, it was difficult to specify the responsible amino acid. These data were explained in the Results as follows: **'These data suggest that five amino**

acid polymorphisms in the RBD specific for the GTB strain were responsible for the attenuated neutralizing effects of L-HBs+Addavax-induced antibodies (Supplementary Fig. 5). These data also indicate that the major antigenic region of L-HBs+Addavax-induced antibodies was in the RBD of the preS1 region. To identify the responsible amino acid, we replaced one of these amino acids with the GTC type and evaluated the neutralizing activities of L-HBs+Addavax-induced antibodies against these strains. The five strains of HBV/NL-RBD-GTB with GTC-type amino acids (-K35G, -E39N, -L45F, -H48N, and -N51H) showed attenuated neutralizing effects at a similar level to HBV/NL-RBD-GTB (Supplementary Fig. 7), and the responsible amino acid was not specified. (page 18, line 314-324), and in the Discussion as follows: 'The various chimeric HBV reporter viruses between GTB and GTC revealed that amino acid polymorphisms of the RBD in the preS1 region are responsible for the attenuated neutralizing activity of L-HBs+Addavax-induced antibodies, although we could not specify the responsible amino acid in this region.' (page 24, lines 421-422).

11) *Since a simple HBV genotype change can abrogate the neutralization potential of L-HBs-specific antibodies, it seems likely that different genotypes of the virus will simply evolve new "VEMs" that are simply the GTB sequence in Pre-S1. Clearly that virus has no fitness disadvantage. Overall, won't it be easier for HBV to evolve to evade your vaccine compared to the contemporary vaccine?*

Response:

We agree that VEMs to the novel L-HBsAg vaccine may emerge. In this paper, we propose this vaccine as a complementary vaccine rather than replacement of the S-HBsAg vaccine used worldwide. The emerged VEMs to L-HBsAg vaccine-induced antibodies will be neutralized by S-HBsAg vaccine induction. Therefore, the combined use of these HB vaccines will be needed and useful. To emphasize this point, we added the following sentence to the Discussion: '**HBV strains of other genotypes or with future emerging VEMs will be neutralized by antibodies induced by either of these vaccines.**' (page 24, lines 433-434).

Reviewer #2 (Remarks to the Author):

Background: The intent of treatment for HBV is usually limited to suppressing viral replication, as complete elimination of the virus can rarely be achieved with current therapies. Therefore, preventative vaccination is considered the best strategy for managing HBV infection. However, current vaccines targeting the S-HBs antigen fail to generate strong humoral responses in 10% of vaccinated individuals, and a small number of vaccine-escape mutations (VEMs) have been reported that evade neutralization.

Noteworthy results: Washizaki et al. developed a complementary vaccine that targets the large HBs antigen, which contains the NTCP binding site. They tested the vaccine in rhesus macaques and in cell culture and reported neutralization activity similar to that of the current HBV vaccine but unaffected by current VEMs.

Significance: The authors do not propose the vaccine as a replacement for the current widely used HBV vaccine but rather as a complementary vaccine intended to overcome various shortcomings and gaps in the protection offered by the standard vaccine. Part of this strategy is to include the NTCP-binding region in the vaccine target, which might improve the efficacy of the vaccine by directly targeting a key functional region of the S protein. The L-HBsAg region is only present on Dane particles, whereas S-HBsAg is also present on the vastly more abundant sub-viral particles, which might serve as a form of immune decoy or sink. However, the current vaccine requires three doses, and the regimen described by the authors also appears to require three doses, so it is unclear whether a six-dose vaccine schedule against essentially the same target is sufficiently practical or necessary. A related approach that achieves many of the same objectives is to target the NTCP receptor instead (e.g., PMID: 34363922), which directly blocks infection by preventing the virus from entering the cell and is less affected by VEMs or differences among viral genotypes.

Response:

We appreciate the reviewer's understanding and deep insight. The combined use of these vaccines is exactly what we aimed to propose. We agree that the administration regimen of these vaccines is an important issue. Currently, we have no data on whether it is better to administer the vaccines simultaneously or sequentially. This would require a future study to be done. The strategy targeting NTCP is attractive, but it is not an HBV vaccine and would be used as another anti-HBV reagent.

Washizaki et al. examined the most common genotypes (A, B, C, and D), which is more than sufficient to support their conclusions, but the study might have greater significance if the same results could be confirmed for some of the less common genotypes and minor strains, especially if the vaccine provides notably better protection than the standard vaccine. Stable HBVcc cell lines producing high titers of HBV genotypes A2, B2, C1, E, F1b and H are available (PMID: 34985994).

Response:

As mentioned by the reviewer, we have already assessed the neutralization of vaccine-induced antibodies to strains of major HBV genotypes. Again, the purpose of this L-HBsAg vaccine is to complement the shortcomings of the standard S-HBs vaccine. The combination use of these vaccines induces antibodies that can neutralize the multiple strains of HBV. It would be difficult to assess many HBV strains, such as HBV with all reported VEMs or minor genotypes by using the HBV reporter virus infection system.

- Does the work support the conclusions and claims, or is additional evidence needed?

The authors' conclusions appear to be sound with sufficient supporting evidence.

- Are there any flaws in the data analysis, interpretation and conclusions

The data analysis does not appear to contain flaws. However, the conclusions feel incomplete without examination of other genotypes and evidence of long-term protection. It is also unclear how and to whom the vaccine would be administered in relation to the standard vaccine. The authors note that the response to the vaccine was carefully monitored in the macaques, but prediction of potential off-target effects in humans should also be considered.

Response:

The long-term protection of antibodies induced by the administration of the L-HBsAg vaccine is still unclear. The protection would depend on the maintenance of induced antibodies. Understanding this question would require another study to establish an effective administrative regimen of these vaccines and to evaluate the maintenance of induced antibodies and their neutralizing effects. For off-target effects, we evaluated the liver dysfunction of vaccinated rhesus macaques

(Supplementary Fig. 1) and found that these vaccines had no side effects on macaques. The safety of these vaccines in humans will be investigated in future studies.

- Is the methodology sound? Does the work meet the expected standards in your field?

The methodology appears to be sound and to meet high standards.

- Is there enough detail provided in the methods for the work to be reproduced?

It should be possible to reproduce the work based on the level of detail provided.

Reviewer #3 (Remarks to the Author):

In this manuscript, Kato et al studied a L-HBsAg-derived vaccine candidate and its efficacy against previously described vaccine escape mutants (VEM) and various HBV genotypes in comparison to standard S-HBsAg vaccine. They showed that the L-HBsAg induced antibody response is more potent against VEM by directing the antibody response mostly to the pre-S1 region instead of the “a” domain of the HBsAg, where the VE mutations reside. On the other hand, it appears to be more variable and less effective against HBV genotypes other than the genotype of the vaccine. This is not surprising given the high level of variations in the pre-S1 region. Previous studies have attempted to generate L-HBsAg-based vaccine on account of the same rationale, but the yield of production and quality of the antigen have been marginal and thus never gain much traction. This study is useful in advancing the goal of developing this type of vaccine, which may have an intrinsic advantage as discussed by the authors. The authors did a nice job in characterizing the induced antibodies in both rhesus and human and clearly defined the different antibody responses between the L-HBsAg and the standard S-HBsAg vaccines. The data are reasonably convincing and the analysis and interpretation are appropriate. Based on this study, it is clear that the L-HBsAg vaccine is more effective against VEMs but less so against other HBV genotypes, which is more of a problem if the aim is to produce a widely effective vaccine. VEMs have not been a serious problem and most of HBV vaccine induced antibodies are sufficiently potent to neutralize many of the VEMs. Thus I am not sure whether this vaccine candidate is any better than what we have. A few other major comments about this study.

Response:

In this paper, we proposed the complementary use of the L-HBsAg vaccine with the standard S-HBs vaccine. The combined use of these vaccines simultaneously or sequentially may be able to induce the antibodies that can neutralize HBV strains with VEMs or multiple genotypes. We found that the IC₅₀ value of antibodies induced by the standard S-HBs vaccine against HBV with VEMs was approximately 3 to 8 times higher than that against wild-type HBV. These data indicate that antibodies induced by this vaccine neutralize HBV with VEMs when the antibody titer is high enough but cannot neutralize them when the antibody titer becomes low. In addition, these data reasonably explain the emergence of HBV with VEMs among the population vaccinated with the S-HBs vaccines described in references 12, 13, and 14.

1. *It is probably more appropriate to compare the standard S-HBs vaccine together with the same adjuvant (K3-SPG or Addavax) against the L-HBsAg + adjuvant. It is clear the adjuvant can significantly enhance the antibody response.*

Response:

As we explained above, the purpose of this paper was not to compare the strengths of the 2 kinds of HB vaccines. As mentioned by the reviewer, Addavax plays a pivotal role in the efficient induction of antibody production by the administration of the L-HBsAg vaccine. Therefore, this adjuvant may be able to contribute to antibody induction even in a population that has a low or no humoral response to the standard S-HBs vaccine observed in approximately 10% of vaccinated adults. To emphasize this point, we added the following sentence to the Discussion: 'The combination of these vaccines that includes different adjuvants may be effective in inducing the production of antibodies even in a population that has a low or no humoral response to the S-HBs vaccine as observed in approximately 10% of vaccinated adults.' (page 24, line 435-437).

2. *Would administration of both S-HBs and L-HBsAg vaccines generate an additive antibody response? It would be of interest to study such a combination.*

Response:

The additional or synergistic effects of the combined administration of these vaccines may be considerable. To this end, the investigation of the vaccination regimen of these vaccines, such as administration timing, concentrations, and the use of simultaneous or sequential administration, will be important. This will be performed as another study in the future.

3. *The recent approval of the Dynavax HBV vaccine (HEPLISAV-B) is formulated with an adjuvant that is a TLR agonist and can induce much higher tiers of anti-HBs, and requires only two doses of vaccine. It would be worthwhile to compare the L-HBsAg based vaccine to this one.*

Response:

HEPLISAV-B has been reported as a novel potent HB vaccine. However, it contains S-HBsAg, which is similar to the standard HB vaccine. This paper discusses the characteristics of the induced antibodies,

including whether they are affected by VEMs or HBV genotypes. The combined use of the L-HBsAg vaccine with HEPLISAV-B may be better than the combination with the current S-HBsAg vaccine, but assessing this would require another study.

Minor points:

1. *In the section on the HBV genotypes, the authors equate neutralizing activity was higher with higher fold of IC₅₀. But the fact is the lower the IC₅₀ the more potent the neutralizing activity. The authors should be careful about the description.*

Response:

For the research area of antiviral reagents, a low IC₅₀ value is recognized as indicating potency. Thus, we followed this theory. Regarding the confused description, mistakes were corrected.

2. *In Suppl Table S1 and S2, the last column is labeled as % of WT, but the number given is a fraction, which should be revised.*

Response:

We have made the suggested corrections.

3. *The Discussion repeats much of Results and should be more concise and focus on the importance and relevance of the data in the context of the current knowledge. The authors provided a less than adequate and unbiased discussion of their study.*

Response:

Following the reviewer's suggestion, we shortened the Discussion section by deleting the redundant explanation (second and third paragraphs in the Discussion). We added several discussions about the outcome of our study following other reviewers' suggestions.

REVIEWERS' COMMENTS

Reviewer #1 (Remarks to the Author):

The authors have responded to my critiques well.